# Eye Involvement in Wilson’s Disease: A Review of the Literature

**DOI:** 10.3390/jcm11092528

**Published:** 2022-04-30

**Authors:** Kevin Chevalier, Martine Mauget-Faÿsse, Vivien Vasseur, Georges Azar, Michaël Alexandre Obadia, Aurélia Poujois

**Affiliations:** 1Department of Neurology, Rothschild Foundation Hospital, 75019 Paris, France; kevinchevalier05@gmail.com (K.C.); aobadia@for.paris (M.A.O.); 2National Reference Center for Wilson’s Disease and Other Copper-Related Rare Diseases, 75019 Paris, France; 3Department of Ophthalmology, Rothschild Foundation Hospital, 75019 Paris, France; mmfaysse@for.paris (M.M.-F.); gazar@for.paris (G.A.); 4Clinical Research Center Coordinator, Rothschild Foundation Hospital, 75019 Paris, France; vvasseur@for.paris

**Keywords:** Wilson’s disease, Kayser–Fleischer ring, eye involvement, sunflower cataract, copper

## Abstract

Wilson’s disease (WD) is an autosomal recessive genetic disorder due to a mutation of the ATP7B gene, resulting in impaired hepatic copper excretion and accumulation in various tissues. Ocular findings are one of the hallmarks of the disease. Many ophthalmological manifestations have been described and new techniques are currently available to improve their diagnosis and to follow their evolution. We have performed a systematic PubMed search to summarize available data of the recent literature on the most frequent ophthalmological disorders associated with WD, and to discuss the newest techniques used for their detection and follow-up during treatment. In total, 49 articles were retained for this review. The most common ocular findings seen in WD patients are Kayser–Fleischer ring (KFR) and sunflower cataracts. Other ocular manifestations may involve retinal tissue, visual systems and eye mobility. Diagnosis and follow-up under decoppering treatment of these ocular findings are generally easily performed with slit-lamp examination (SLE). However, new techniques are available for the precocious detection of ocular findings due to WD and may be of great value for non-experimented ophthalmologists and non-ophthalmologists practitioners. Among those techniques, anterior segment optical coherence tomography (AS-OCT) and Scheimpflug imaging are discussed.

## 1. Introduction

Wilson’s disease (WD), also known as hepatolenticular degeneration, is a rare genetic condition due to a recessive mutation of the ATP7B gene. The disease consists of a continuous copper accumulation in many tissues and requires life-long treatment. With fewer than 1000 cases in France [1], this condition is mainly characterized by hepatic, ophthalmological and neurological features due to copper accumulation in those organs. Treatments available for this disease rely mainly on copper chelators (D-Penicillamine and trientine salts) and zinc salts.

The ophthalmological manifestations are one of the hallmarks of the disease. Corneal deposition of copper, called Kayser–Fleischer ring (KFR), is typical in WD and constitutes one of the diagnostic criteria for this disease [2]. A sign of extra-hepatic copper accumulation, KFR is a useful biomarker that allows the evolution under chelator treatments. It is a reversible sign and may disappear under treatment. Apart from KFR, many other ophthalmological manifestations have been described. In addition, new techniques have made it possible to improve the diagnosis of these manifestations and to follow their evolution [1]. In this report, we conducted a systematic review of the recent literature in order to identify the ophthalmological disorders associated with WD, as well as their prognosis and evolution under treatment. Moreover, we also describe new techniques used for their detection and follow-up.

## 2. Materials and Methods

A systematic literature review was performed on PubMed to identify case reports and studies that treat WD and its eye involvement. The literature search was conducted systematically following the Preferred Reporting Items for Systematic Reviews and Meta-Analyses (PRISMA) guidelines (http://www.prisma-statement.org, accessed on 1 February 2022). In order to capture the newest data and technics, only studies that were published between January 2000 and January 2022 were searched for. The following search terms without any search filters were used: Wilson disease AND Eye NOT Wilson [Author]. Only publications in English and French were retained. Articles treating both adult and pediatric WD patients were included, whereas those reporting only images with descriptions were excluded. The selection of the relevant articles was made by considering the titles, then the abstracts. All the titles and abstracts were screened. Reports not dealing with WD and/or the ophthalmologic system were excluded. Precedent reviews of the literature and their related articles were included. This review was based on full-text articles only.

## 3. Results

In the first step, 122 publications were identified on PubMed and 73 of them were excluded, either because of not mentioning WD (*N* = 14), not concerning the ophthalmological system (*N* = 31), being written in a language other than French or English (*N* = 10) or being based on only images with their description (*N* = 15). The full text access was unavailable in three publications. Finally, 49 publications were included in this review (Figure 1).

### 3.1. Cornea Involvement

KFR was first described in 1902 and 1903, respectively, by two German ophthalmologists, Benhard Kayser and Bruno Fleischer. It consists of a ring-shaped copper deposit in the anterior chamber angle within the internal corneal layer of the Descemet’s membrane, at the Schwalbe’s line (Figure 2) [3,4]. In fact, copper is deposited throughout the cornea, and sulfur-copper complexes producing the visible copper deposits are formed only in Descemet’s membrane [5]. On SLE, KFR appears like a golden-brown, golden-green, green-yellow, golden-yellow, bronze or reddish-brown coloring ring in the limbic area of the cornea [6]. It first develops in the superior part of the cornea (at the 12-o’clock position), then inferiorly, and finally in the horizontal meridian forming a closed ring. [3,7,8,9] Thus a closed KFR is evidence of a long-term disease [3]. This pattern of formation and resolution of the KFR could be explained by the vertical flow of aqueous fluid in the anterior chamber of the eye [10]. It is usually bilateral [3] but a unilateral case was reported in 1986 [11].

KFR confirms the presence of excess free copper in the bloodstream but is not pathognomonic for WD, as it may occur in any disorder with impaired biliary copper excretion [7,12,13,14,15,16]. KFR is not constantly detected by the classical SLE in WD; its prevalence is estimated between 36–62% in patients with hepatic manifestation, between 77.8–85.2% in patients with neurological manifestation [3,5,17] and between 10–30% in asymptomatic WD patients [3,4,5,9,17,18,19,20,21,22,23]. The incidence of KFR also varies according to the age of the diagnosis. Indeed, the largest pediatric cohort in WD has been recently published by Couchonnal et al. describing 182 children with WD. In this cohort, at diagnosis, 149 (81.8%) children had an ophthalmologic evaluation [24]. Among them, 58 (38.9%) had a detectable KFR: 40/129 (31.0%) were hepatic patients and 18/19 (94.7%) were neurological patients. The youngest patient with a detectable KFR was a seven-year-old, and a total of eight patients (13.7%) with detectable KFR were younger than 10 years, all were hepatic patients. The incidence of KFR in children is much lower than in adults. Nevertheless, it is puzzling that KFR is more frequent in neurological patients (like in adults) but more early in hepatic patients. To compare, in one of the largest adult cohorts from Merle et al. (163 patients), KFR was detected in 66.3% of the patients and more frequently in those with neurological symptoms than those with hepatic symptoms (85.5% vs. 52.1%, *p* < 0.001) [25] (Table 1).

In vivo confocal microscopy (IVCM) shows that KFR consists of granular, bright particles that increase in density toward Descemet’s membrane and is associated with a decreased number of keratocytes and peculiar dark, and round areas in all stromal layers. When the ring is not visible in subjects with WD, changes to the corneal microstructure are insignificant [26]. A significant statistical correlation between the presence of KFR, and the clinical neurological manifestations, brain magnetic resonance imaging (MRI) or the toxic free copper (so-called exchangeable copper) was found in some studies [6,27]. On the other hand, the absence of KFR in neurologic WD patients is correlated with higher ceruloplasmin concentrations and serum copper levels, less liver cirrhosis and less prominent signal changes in brain MRI than in patients with KFR [18].

The presence of KFR, identified with SLE by a skilled examiner or rarely with the naked eye, is included in the current diagnosis criteria of WD [2,28,29]. KFR usually disappears progressively with effective treatment of WD, fading initially from lateral and medial aspects of the cornea, then finally from its superior part. As its recurrence suggests a non-adherence to treatment, close monitoring of the ocular status during the follow-up is highly recommended [3,6,9,23,26,28,30,31,32,33,34]. Nevertheless, its disappearance is not correlated with the resolution of the other signs or symptoms of the disease [3,6,8,14].

The sensitivity (Se) of SLE to detect a KFR is known to be low (KFR was missed in more than 50% of hepatic WD patients) [2,35]. That could be explained by the predominance of copper deposits in the anterior chamber angle that cannot be detected with a standard SLE. Therefore, a thorough ophthalmological exam with gonioscopy, which permits a detailed examination of the iridocorneal angle structures, remains crucial [3]. SLE and gonioscopy both required experiment operators, as non-trained ophthalmologists could miss the KFR [3]. These findings lead to the proposal of other methods that could be used to assess the presence and progression of copper deposits in the cornea.

Anterior segment optical coherence tomography (AS-OCT) could be used for the detection of KFR. The KFR appears on a grey scale as a hyper-reflective layer at the level of Descemet’s membrane in the peripheral cornea (Figure 3). On a color scale, it appears as a green/green-yellow/yellow/yellow-orange band. KFR can be easily measured using the gray scale of AS-OCT [3,30,31]. In a study of 29 patients with WD, 15 had normal slit-lamp evaluation but abnormal AS-OCT (*p* < 0.001) hypothesizing that AS-OCT is a more accurate diagnostic tool that could detect significantly more cases of KFR as compared to the slit-lamp evaluation in participants with hepatic and neurological WD manifestations [3]. This technique could permit more easy recognition of KFR for non-experimented ophthalmologists, as well as non-ophthalmologists practitioners [30,32]. Moreover, it is also useful in children and non-cooperative patients because the imaging process involves fixation for only a few seconds without exposure to bright light [36]. AS-OCT could possibly determine the density of copper deposit in KFR and help the clinician determine the severity of the disease. Further studies are needed to know if AS-OCT can distinguish KFR from pigmented corneal rings in non-Wilsonian liver disease or arcus senilis and if repetitive AS-OCT could help assess the good response of chelator treatments in WD [30].

Pentacam HR Scheimpflug imaging is a device that can provide three-dimensional image representations of the anterior segment, which may be useful for screening narrow angles. Its application in WD was discussed because of the predominance of copper deposits in the anterior chamber angle. In Scheimpflug imaging, KFR could be seen as a bright subendothelial band at the periphery of the cornea. In a study of 21 WD patients (mean age 33 years), patients with KFR (*n* = 11) had a significantly higher subendothelial signal than patients without KFR (*n* = 10) or healthy controls (*n* = 9) (*p* < 0.05) [33]. An extension of this study was published three years later by the same team [37]. In the study, 10 patients with KFR were compared to five WD patients with other causes of peripheral corneal scatter, 16 WD patients with normal ophthalmologic examination and 10 healthy controls. Scheimpflug images of the posterior 60 μm and anterior 120 μm of the peripheral cornea were compared to SLE as a gold standard. Scheimpflug images were exported in ImageJ (NIH, Bethesda, MD, USA) (a software for image analysis) to determine a profile of signal intensity (0–256). Using ImageJ, a calculation of a normalized signal ratio (peak posterior/peak anterior value) was created and showed that a ratio > 1 had a high sensitivity (96%) and specificity (95%) for KFR detection. Follow-up and multiple examinations of 12 patients of the same cohort suggested that this method may be used to follow-up the patients as a marker of whole-body copper reduction under treatment. In a recent study by Degirmenci and Palamar, Scheimpflug image was used to diagnose and make a grading of KFR in 22 WD patients compared to controls [38]. KFR was observed at the periphery of the cornea as a hyper-reflective area, the extent of which was concordant with KFR severity on SLE. Moreover, the corneal volume, the central corneal thickness, the corneal thickness at 2, 4, 6, 8 and 10 mm measured with Scheimpflug image were positively correlated with the urinary copper/24 h (*p* < 0.001, 0.023, 0.006, 0.004, 0.001, <0.001 and <0.001, respectively) at diagnosis. As previously discussed with AS-OCT, this method could also be a useful diagnostic tool for non-ophthalmologists to detect KFR in WD patients.

In vivo confocal microscopy by laser scanning confocal microscopy (LSCM) was evaluated by Ceresara et al. for the diagnosis of KFR [35]. Twenty patients with WD (mean age 36.6 ± 10.4 years, 13 hepatic forms and 7 neurologic forms) who were under treatment (chelators (*n* = 13), zinc salt (*n* = 7)) and 20 age- and gender-matched controls (mean age 37.2 ± 8.3 years) were compared. In this cohort, SLE showed KFR in five WD patients (25%) and LSCM showed peripheral hyperreflective granular microdeposits at the level of Descemet’s membrane in 15 WD patients (75%) but not in healthy controls. LSCM showed that the intensity of the hyperreflective granular confocal deposits was proportional to the size of the pigmented area of KFR (*p* < 0.001). On the other hand, to assess the accuracy of IVCM in detecting KFR, Zhao et al. compared 52 WD patients with KFR (mean age 24.8 years) with 52 healthy controls (mean age 25.0 years) [34]. They found that all patients with KFR had an abnormal IVCM, which suggests the interest of this technique in KFR detection. Nevertheless, the daily use of LSCM is still limited by its price on one hand, and the necessity of thorough training to interpret properly the imaging results on the other hand [32].

Corneal densitometry is a parameter related to the transparency of the cornea. This parameter was assessed with Pentacam high resolution (PHR) in 20 WD patients without KFR and 18 with KFR and compared to 18 healthy individuals [13]. Patients with WD and KFR had significantly increased center and posterior corneal densitometry values in the peripheral paracentral cornea. In addition, the corneal thickness and volume values were significantly lower in WD patients with and without a KFR than in healthy individuals. Further studies are needed to assess whether corneal densitometry with PHR could be used for diagnosis and treatment monitoring in patients with WD.

Other methods like X-ray excitation spectrometry have been tried. However, despite changes in corneal copper content in patients without KFR in SLE, this instrument is not widely available and exposes patients to a potential risk of lens irradiation [32,39].

Aside from KFR, corneal nerve fibers are also impacted by copper deposits. A comparative study between 24 WD patients (mean age 35.1 ± 8.2 years) and 24 healthy matched-subjects (mean age 35.3 ± 9.3 years) showed, in corneal confocal microscopy, a significant difference in superficial epithelium cell diameter (2.49 ± 1.70 μm vs. 20.78 ± 2.62 μm, *p* < 0.0001) and basal epithelium cell density (4260 ± 442.10 cell/mm^2^ vs. 6885 ± 265.10 cell/mm^2^, *p* < 0.0001). All parameters of the sub-basal nerve plexus were significantly different between the two groups: nerve fiber length density (*p* < 0.0001), number of fibers (*p* < 0.001), number of beadings (*p* = 0.025), and number of branchings (*p* < 0.0001) were significantly lower, whereas fiber tortuosity was significantly higher (*p* < 0.0001) in WD subjects versus controls [40].

### 3.2. Lens Involvement

Sunflower cataract is another classical ocular manifestation of WD, the frequency of which varies widely in the literature (between 2 and 20%) [10]. A recent study performed by Langwinska-Wosko et al. on 81 consecutive newly diagnosed WD patients reported that such cataract was detected in only one (1.2%) of all patients, suggesting that sunflower cataract is a very rare ocular sign of WD in de novo and untreated patients. After a year of treatment, the cataract fully disappeared [17]. This manifestation was first described by Siemerling and Oloff in 1922 as “cataracts like rays of the sun” [41]. The authors already noted the similarities between the cataract seen in their patients with WD to the one produced by an intraocular foreign body containing copper [9]. Sunflower cataract consists of copper deposition in the lens capsule and not within the lens cortex or nucleus itself, and has an aspect of a central disk with radiating petal-like spokes that give its name [7]. A Korean team has studied the elemental composition of the lens capsule in sunflower cataracts thanks to transmission electron microscopy (TEM), with energy-dispersive X-ray spectroscopy (EDS) in a capsulorrhexized anterior lens capsule in a 37-year-old male WD patient [42]. TEM showed the presence of granular deposits mainly in the posterior one-third of the capsule. EDS found consistent peaks for copper and sulfur in all of these electron-dense granules. Sulfur is present in copper-binding proteins such as metallothioneins explaining its presence in the lens. Sunflower cataract seems to be the result of the accumulation of heterogeneous compounds in the third posterior of the lens’ anterior capsule including copper, sulfur and/or binding-copper proteins.

Sunflower cataracts usually do not impair vision, cannot be seen with the unaided eye or with an ophthalmoscope, and require slit-lamp evaluation for detection [23]. Like KFR, sunflower cataracts usually regress with copper-chelating treatment [17,43].

### 3.3. Macula, Retinal Nerve Fiber Layer and Visual Pathways Involvement

Five studies have investigated the relationship between retinal and visual pathways damage and brain involvement associated with WD. Correlation between brain MRI lesions and impairment of visual pathways, macula and retinal nerve fiber layer (RNFL) was found by Langwinska-Wosko et al. [44]. They compared 58 WD patients mean age 38.7 years, with or without brain lesions on MRI (39 MRI+ and 19 MRI−, respectively) and 30 healthy controls (mean age 39.6 years). Total RNFL measured spectral-domain optical coherence tomography (SD-OCT) was thinner in WD patients MRI+ than WD patients MRI− (*p* = 0.001). Central macular thickness (CMT) was also significantly thinner in WD patients MRI+ than WD patients MRI− (*p* < 0.001). No significant difference was found in RNFL or CMT between WD patients MRI− and controls. Latency of visual evoked potentials (PEV) and electroretinography (ERG) were prolonged in WD patients MRI+ compared to WD patients MRI− (*p* < 0.001 and *p* < 0.001, respectively). Interestingly, some WD patients MRI− had electrophysiological abnormalities. These results confirmed what had been already demonstrated by a German team in 2012 [45]. An Indian team has also shown prolonged latencies in PEV and ERG in WD patients with neurological manifestations compared to controls and the improvement of PEV and ERG latencies after treatment of WD [46]. Another study by Langwinska-Wosko et al., showed a significant negative correlation between RNFL, CMT and neurological symptoms (*p* = 0.008 and *p* = 0.04, respectively) [47]. Finally, a recent study by Svetel et al. confirmed that WD patients have a lower RNFL thickness than healthy controls [48]. However, they did not show a correlation between the clinical features of the disease (age, duration of disease, duration of treatment, dosage of D-Penicillamine or neurological assessment) and retinal thickness parameters. In the same way, they did not find any statistical difference in the RNFL thickness between WD patients with neurological or hepatological forms.

OCT, ERG and PEV are potential clinical tools for the investigation and assessment of neurological involvement in WD (Figure 4). However, to date, no studies have shown a correlation between these lesions and the evolution of the disease.

### 3.4. Eye Mobility

WD is responsible for eye movement abnormalities such as slow horizontal and vertical saccades [23,49,50], abnormal vertical smooth pursuit [51], increased antisaccadic latency and error rate [52]. Ingster-Moati et al. found that 91% of 34 WD patients (mean age 29 years, 24 neurological forms, 9 hepatic forms and 1 asymptomatic patient) had abnormalities of ocular motility detected by electro-oculography. Here, 29 patients (85%) had an abnormal vertical smooth pursuit, 41% a vertical optokinetic nystagmus and 41% an impaired horizontal smooth pursuit. Among the 27 who underwent MRI, seven patients had normal brainstem and lenticular nuclei images despite the detection of ocular motility abnormalities [51]. In a study comparing 20 WD patients (mean age 46.8 years) to 20 age- and sex-matched controls (mean age 46.4 years), WD patients showed prolonged latencies of horizontal prosaccades (t = 2.3, *p* = 0.03) and decreased gain (= hypometry) of both horizontal (t = −2.2, *p* = 0.04) and vertical (t = −2.1, *p* = 0.046) prosaccades [53]. They also had prolonged latency of both horizontal (t = 2.2, *p* = 0.04) and vertical (t = 2.1, *p* = 0.047) antisaccades and increased error rate of vertical antisaccades (t = 2.2, *p* = 0.04) compared to controls. Brain MRI of WD patients showed a strong association between prolonged latencies of prosaccades and the brainstem atrophy (r = −0.53 and *p* = 0,02 for horizontal latencies and r = 0.47 and *p* = 0.004 for vertical maximum speed in prosaccades, respectively). Impairment in eye movement is probably secondary to the lesions induced by the copper deposit in the brainstem as the nerve centers responsible for vertical and horizontal eye tracking are located in the midbrain and the pons, respectively.

Involvement of basal ganglia could lead to extrapyramidal-like disorders. Verma et al. reported the case of a 24-year-old woman with WD who presented Parkinson-like symptoms and increased blinking rates at 32/min. MRI showed T2-weighted hypersignal in the basal ganglia and the brainstem. Her blinking rate normalized rapidly in a month after L-Dopa and decoppering treatment. Authors hypothesized that the increase in blinking rate was due to depletion of nigrostriatal dopamine, which led to hyperexcitability of blink reflex [54].

Rapid eye movement (REM) sleep behavior disorder is also described in WD. In a study by Tribl et al., four patients with WD and REM sleep behavior disorder were described, three of which presented with REM sleep behavior disorder as the first initial symptom of the disease. All patients showed mesencephalic or ponto-mesencephalic lesions in brain imagery. Moreover, in a recent study, sleep complaints and disease symptoms were compared in 40 patients with WD (20 patients with hepatic phenotype matched to 20 neurologic ones) and 40 age-, sex- and body mass index (BMI)-matched healthy controls [55]. REM sleep behavior disorder was more frequent in WD than in the controls (20% vs. 0%, respectively, *p* = 0.005) and it had started two years before the diagnosis of WD in one patient and after (from 10 to 29 years) in the others. Some patients presented with frequent (more than twice per week in six patients) or aggressive episodes (two patients). Interestingly, the frequency of REM sleep behavior did not significantly differ between WD patients with neurologic or hepatic forms. Like in Parkinson’s disease, REM sleep behavior disorder could be one of the first symptoms of neurological WD and its detection could permit an earlier treatment. Moreover, the decoppering therapies could treat REM sleep behavior in patients with WD, reducing the copper overload in the subcoeruleus region responsible for REM [55].

### 3.5. Other Ocular Abnormalities

Other ophthalmological involvements have been described, essentially in case reports: night blindness, exotropic strabismus, optic neuritis, optic disc pallor, rhegmatogenous retinal detachment, loss of accommodation response, and eyelid opening apraxia [7,56,57]. In fact, few case reports have described the association between WD and optic neuropathy with optic disc pallor. In three clinical cases, extensive investigations did not find any other cause for the optic neuropathy and the best-corrected visual acuity (BCVA) improved after treatment of WD [58,59,60]. Nevertheless, no case series have been published yet to assert this association.

A case of glaucoma was reported in an 18-yo patient who presented with both forms (hepatic and neurologic form) of WD [61]. BCVA was reduced in the right eye (OD, hand motion) and normal VA in the left eye (OS). Intraocular pressure was 44 mmHg OD and 15 mmHg OS. Gonioscopy revealed bilaterally wide and opened iridocorneal angles, but a thick layer of yellow-grey material covered the trabecular meshwork OD. A biopsy from the trabecular meshwork exhibited copper ion staining.

Finally, two cases of WD patients with keratoconus at presentation were described in the literature [62,63]. In these two cases, keratoconus was diagnosed simultaneously with a KFR in asymptomatic WD patients. Nevertheless, the association between keratoconus and WD is not certain and could be a pure coincidence.

## 4. Discussion

Ocular manifestations may be the first presenting symptoms of WD, which must be recognized to prevent fatal outcomes. To the best of our knowledge, this is the first systematic recent review of the literature with regard to ophthalmological involvement in WD. Moreover, we also attempt to describe new techniques that may be useful in the early detection and follow-up of the disease.

As KFR is an essential criterion for the diagnosis of WD, multiple methods have been studied to improve its diagnosis. So far, SLE is the gold standard for detection of KFR, but its Se is low and it requires experimented ophthalmologists. Since 2016, AS-OCT studies demonstrated a better Se and Sp to diagnose KFR compared to SLE. Its interpretation appears easier and more accessible for non-experimented ophthalmologists and non-ophthalmologist practitioners [30,32]. The density of copper deposit in the cornea at the diagnosis and during the follow-up of WD could determine the severity of the disease and the response under chelator treatment [30]. Pentacam HR Scheimpflug imaging using ImageJ software and calculation of the ratio between anterior and posterior peak signal presents also a good Se (96%) and specificity (Sp, 95%) to detect KFR. Nevertheless, the utilization of those methods to assess therapeutic efficacy needs further evaluation.

In addition to KFR, the evaluation of anterior segment in WD patients showed differences with healthy population. In a study published by Kara et al., 22 patients with WD (mean age 29 years old) were compared to 22 age- and sex-matched controls [64]. WD patients had significantly higher CCT (*p* < 0.001), keratometric values (*p* = 0.011), anterior and posterior corneal elevation (*p* = 0.029 and *p* = 0.002, respectively), horizontal visible iris diameter (*p* = 0.001) and anterior chamber depth (*p* = 0.01) compared to controls.

In the pediatric population, the cornea and the lens clarity were also impacted. In the study of Doguizi et al., 24 WD patients (mean age 13.4 ± 3.8 years, mean duration of the disease 4.8 ± 3.4 years) were compared to 25 controls (mean age 13.3 ± 3.0 years) [65]. Scheimpflug tomography and PHR were used to measure the changes in corneal and lens transparency. In WD eyes, the corneal densitometry values were higher in the posterior total diameter (*p* < 0.037) and the total thickness of 10–12 mm (*p* < 0.032), than in control eyes. Average and maximum lens densitometry values were significantly higher in WD patients compared to controls (*p* < 0.001 and *p* = 0.006, respectively). These values were significantly correlated with the duration of the disease and the liver copper content (*p* = 0.012 for corneal densitometry and *p* = 0.018 for lens densitometry, respectively).

With regard to eye movement impairment, the association between eye movement abnormalities and brainstem atrophy has been demonstrated by Hanuska et al. [53]. They suggest that video-oculography could be a sensitive electrophysiological marker of brainstem dysfunction in WD patients. Furthermore, abnormal ocular motility may be observed even in the absence of abnormalities in brain MRI. Indeed, in a study by Ingster-Moati et al., 7 out of 27 patients with abnormal ocular motility had a normal brain MRI [51].

VA are usually preserved in patients with WD. Except in the case of late diagnosis, the presence of sunflower cataracts appears to have a limited effect on patients’ VA [17,66]. This should be emphasized, as classical cataracts usually have a marked effect on vision. Sunflower cataract is not a “true” cataract, as it is caused by reversible copper deposition under the anterior capsule of the lens. Indeed, one case was described in the literature by Goyanl and Tripathi in a 10-year-old girl with progressive visual loss for 2.5 years. She had a bilateral KFR and sunflower cataracts [66,67] suggesting an advanced disease. Moreover, neurological manifestations in WD develop due to the deposition of copper in different brain areas like basal ganglia, cerebral cortex, corticospinal and corticobulbar pathways. Despite the rare description of blinding secondary to optic nerve involvement [59,60], visual impairment of neurological origin is very rare and is generally due to delayed diagnosis (like in the two cases reports cited below) or diagnosis error.

As illustrated by the importance of the ocular findings in the diagnosis score for adults [2,68] and pediatric WD [29], the contribution of ophthalmologists is essential in the diagnosis of WD.

With regard to the rarity of the disease and the crucial need for an early diagnosis, access to accessible ophthalmological techniques could improve the diagnosis of WD and, consequently, an earlier treatment improving the prognosis. AS-OCT and other techniques like RNFL measurement are easily accessible, even in an ophthalmology practice outside the hospital. The association of a better sensitivity, specificity, accessibility and easy reading could lead to the recommendation to perform systematically AS-OCT scans and in vivo confocal corneal microscopy in all WD patients with neurologic, psychiatric or hepatic symptoms. These techniques could provide specific findings that could detect earlier the presence of a KFR and RNFL defect. Moreover, the images captured by these techniques do not require the expertise of ophthalmologists and can be easily delegated to experienced orthoptists. The development of telemedicine is another progress that could help the diagnosis of WD. Unlike SLE, these examinations are not operator-dependent, and the images can be sent to an experienced ophthalmologist improving the speed and sensitivity of the diagnosis.

Finally, the development of artificial intelligence (AI) programs such as anterior segment image analysis could be a huge step in the diagnosis of WD. AI will probably be very performant in the detection of KFR since it analyzes grey-scale images in AS-OCT. Indeed, newest programs of AI dedicated to the analysis of the anterior segment by AS-OCT, notably the cornea, are in development [69,70,71]. The difficulty is to recruit and perform this examination on a sufficient number of patients to train the AI with a good performance. The other challenge would be the realization of good images since some patients have neurologic movements making their achievement difficult. In summary, the association of more sensitive non-dependent operator techniques and the development of AI could drastically improve the diagnosis of WD reducing the diagnostic time.

## 5. Conclusions

From this review of the literature, it appears that ophthalmological involvement is frequent in WD patients, in particular for the KFR and, to a lesser extent, sunflower cataracts. Other manifestations involving retinal and visual systems, eye mobility or other structures of the eye have been described with various frequencies. AV is nearly often preserved despite corneal or neurological involvement. The evolution of ophthalmologic manifestations seems to be correlated with decoppering treatment, especially for KFR and sunflower cataracts. New methods like AS-OCT and Scheimpflug imaging are alternatives to traditional SLE. These methods allow non-ophthalmologists to look for and quantify KFR more easily and are useful tools to follow the evolution of these abnormalities under chelating treatment. In the near future, the recent development of AI in the analysis of ophthalmic imaging will probably be helpful for the screening of WD anomalies.

## Figures and Tables

**Figure 1 jcm-11-02528-f001:**
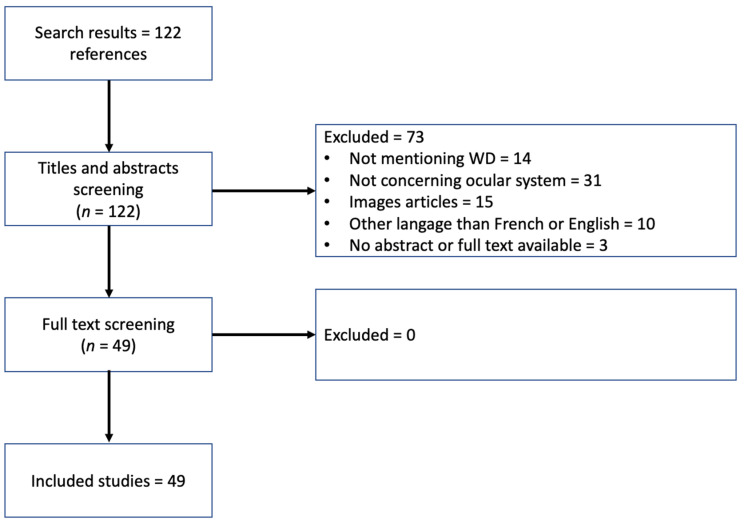
Flow-chart of the literature review. WD: Wilson’s Disease.

**Figure 2 jcm-11-02528-f002:**
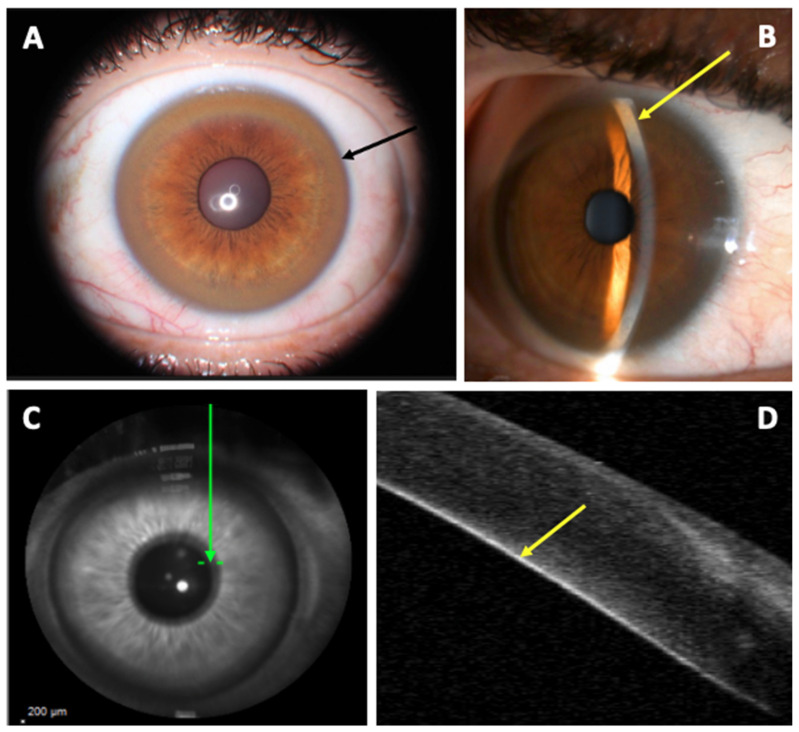
Kayser–Fleischer Ring. (**A**): Slip-lamp examination showing a diffuse circumferential Kayser–Fleischer ring in the left eye (black arrow); (**B**): Slit-lamp examination: visualization of the copper deposit at the posterior part of the cornea in fine slit (yellow arrow); (**C**): Corneal B-scan localization (Spectralis; Heildelberg Engineering) (green arrow); (**D**): marked hyperreflectivity of the posterior part of the cornea corresponding to the copper deposit (yellow arrow).

**Figure 3 jcm-11-02528-f003:**
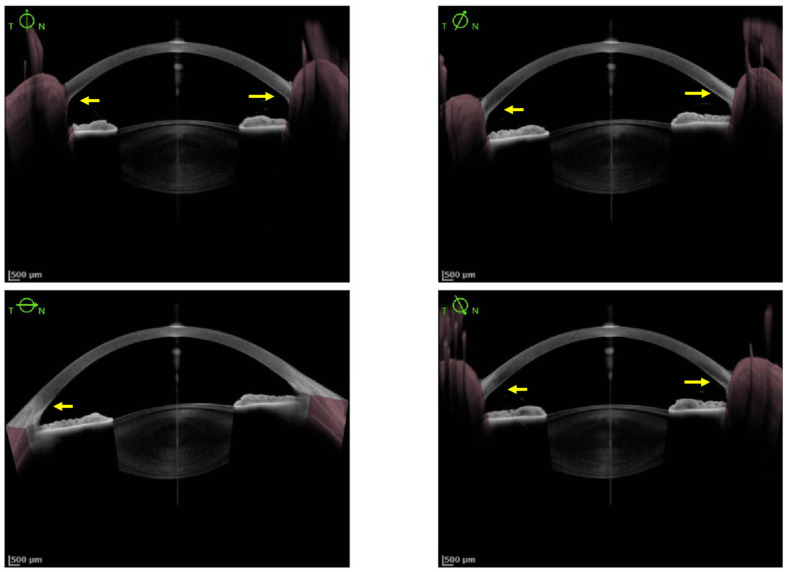
High-resolution Swept Source technology (Anterion^®^, Heidelberg Engineering, Heidelberg, Germany) provides images of very high quality allowing the detection of a faint Kayser–Fleischer ring (KFR) hardly visible on slit-lamp examination (yellow arrow).

**Figure 4 jcm-11-02528-f004:**
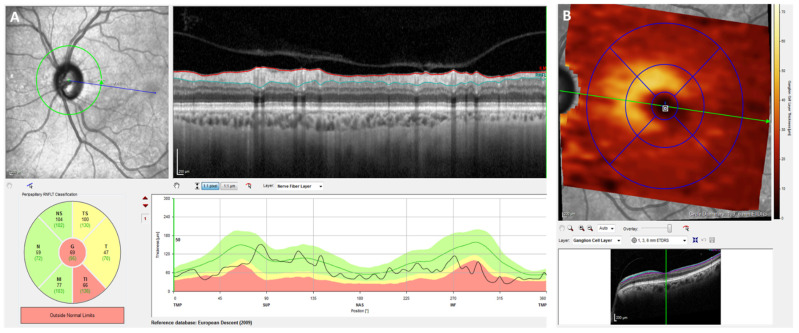
Optic nerve OCT analysis. (**A**): Retinal nerve fiber layer (RNFL) analysis showing optic fiber damage, confirmed on the (**B**) scan image showing particularly an important decrease of the ganglion cell layer (GCL) thickness (yellow part).

**Table 1 jcm-11-02528-t001:** Incidence of Kayser–Fleisher Ring (KFR) according to the form of the disease and the population.

	Pediatric Population [24]	Adult Population [25]
Patients undergoing SLE	149	163
Neurological patients	19	55
Hepatic patients	129	96
Incidence of KFR	58 (38.9%)	108 (66.3%)
Neurological patients	18 (94.7%)	47 (85.5%)
Hepatic patients	40 (31.0%)	50 (52.1%)

KFR: Kayser–Fleisher Ring, SLE: Slit-Lamp Examination.

## Data Availability

Not applicable.

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
