# Peer review of "Eye Involvement in Wilson’s Disease: A Review of the Literature"

_jcm, 2022, doi:10.3390/jcm11092528_

Round 1

Reviewer 1 Report

1.wilson's Disease (WD) eye performance has characteristic, has been found for many years, this review does not refine more clear new intentions.

2.WD eye performance needs to be authenticated.

3. The mechanism of WD eye performance needs to be supplemented.

4. How to improve WD eye diagnosis new techniques is unclear.

Author Response

Dear Reviewer 1,

First, we would like to thank you for the time and effort that you dedicated with your feedback on our manuscript. We are grateful for the insightful comments that add valuable improvements to our paper. We have incorporated most of your suggestions. Those changes are highlighted within the manuscript. Please see below, in blue, a point-by-point response to your comments.

  1. Wilson’s Disease eye performance has characteristic, has been found for many years, this review doesn’t not refine clear new intentions.

We totally agree with the fact that WD has been found for many years. However, the latest review regarding WD’s ocular manifestations goes back to 1997 (Becker et al, Ophthalmologe) and was only available in German language. Our main purpose was to describe a current and updated systematic review of the most recent literature in order to identify ophthalmological disorders, and specially describe those newest techniques that did not exist back that time, and that could be of great value for non-experienced ophthalmologists and non-ophthalmologists practitioners in the diagnosis of ocular manifestations of the disease.

  1. WD eye performance needs to be authenticated & 3- The mechanism of WD eye performance needs to be supplemented.

We thank you again for this insightful comment. As per your request, please find some details about eye performance that we also added in the discussion.  We changed this section, moving and modifying the sentence “Usually, sunflower cataract does not impact the VA but Goyanl and Tripathi have described the case of a 10-year-old girl with progressive visual loss for 2.5 years. She had a bilateral KFR and sunflower cataracts [64,65]” and added some details. You can find below the modified section, in the discussion:

VA are usually preserved in patients with WD. Except in case of late diagnosis, the presence of sunflower cataract appears to have a limited effect on patients’ VA [17,66]. This should be emphasized, as classical cataracts usually have a marked effect on vision. Sunflower cataract is not a “true” cataract, as it is caused by reversible copper deposition under the anterior capsule of the lens. Indeed, one case was described in literature by Goyanl and Tripathi in a 10-year-old girl with progressive visual loss for 2.5 years. She had a bilateral KFR and sunflower cataracts [66,67] suggesting an advanced disease. Moreover, neurological manifestations in WD develop due to deposition of copper in different brain areas like basal ganglia, cerebral cortex, corticospinal and corticobulbar pathway. Despite rare description of blinding secondary to optic nerve involvement [59,60], visual impairment of neurological origin is very rare and is generally due to delayed diagnosis (like in the 2 cases reports cited below) or diagnosis error”

In conclusion, we added a sentence for WD eye performance: “AV is nearly often preserved despite corneal or neurological involvement.”

References added are the following ones:

[17] LangwiÅ„ska-WoÅ›ko, E.; Litwin, T.; Dzieżyc, K.; CzÅ‚onkowska, A. The Sunflower Cataract in Wilson’s Disease: Pathognomonic Sign or Rare Finding? Acta Neurol Belg 2016, 116, 325–328, doi:10.1007/s13760-015-0566-1.

[66] Goyal, V. Sunflower Cataract in Wilson’s Disease. Journal of Neurology, Neurosurgery & Psychiatry 2000, 69, 133–133, doi:10.1136/jnnp.69.1.133.

[67] Negahban, K.; Chern, K. Cataracts Associated with Systemic Disorders and Syndromes: Current Opinion in Ophthalmology 2002, 13, 419–422, doi:10.1097/00055735-200212000-00013.

[59] Gow, P.J.; Peacock, S.E.; Chapman, R.W. Wilson’s Disease Presenting with Rapidly Progressive Visual Loss: Another Neurologic Manifestation of Wilson’s Disease? J Gastroenterol Hepatol 2001, 16, 699–701, doi:10.1046/j.1440-1746.2001.02381.x.

[60] Rukunuzzaman, M.; Karim, M.B.; Rahman, M.M.; Islam, M.S.; Mazumder, M.W. Wilson’s Disease in Children with Blindness: An Atypical Presentation. Mymensingh Med J 2013, 22, 176–179.

  1. How to improve WD eye diagnosis new techniques is unclear.

Thank you for this comment. In an era of very rapid technological progress, it is important to take stock about WD insofar as imaging plays an important role in the diagnosis and follow-up of patients. Since clinical presentations of Wilson’s disease are not specific and are highly variable, it is important -even if this disease is rare- to review regularly the tests used for this disease, the diagnosis of which is still unfortunately often delayed with non-negligible consequences for the patient. Moreover, some of these new techniques described in our manuscript could permit a diagnosis of WD characteristic eye patterns, like the Kayser-Fleischer ring for non-experimented ophthalmologists and non-ophthalmologists’ practitioners.

Moreover, to the best of our knowledge, this current literature review is the first systematic recent review of the literature about ophthalmological involvement in WD covering all the ocular pattern of this rare disease.

To better emphasize these points, we added the following to the discussion part:

“As illustrated by the importance of the ocular findings in diagnosis score for adult [2,68] and pediatric WD [29], the contribution of the ophthalmologist is essential in the diagnosis of WD. 

In regard of the rarity of this disease but the need for an early diagnosis, the access to accessible ophthalmologic technics could improve the diagnosis of WD and, so, an earlier treatment improving the prognosis. AS-OCT and other technics like RNFL measurement are easily accessible, even in an ophthalmology practice outside hospital. The association of a better sensitivity, specificity, accessibility and easy reading could lead to the recommendation of performing systematically, in addition to SLE, on all patients, with neurologic, psychiatric or hepatic symptoms AS-OCT scans or in vivo confocal corneal microscopy both technics providing characteristic images in order to detect the presence of a KFR and also RNFL OCT of the optic nerve, to detect a defect of the optic nerve fiber layer. Moreover, the images capture by these technics do not require the expertise of an ophthalmologist but an orthoptist and can be delegated to an experienced orthoptist. The development of telemedicine is another progress that could help the diagnosis of WD: as these examinations are not operator dependent, unlike SLE, the images could be send to an experienced ophthalmologist improving the speed and sensitivity of the diagnosis. Finally, the development of artificial intelligence (AI) programs such anterior segment images analysis could be a huge step in the diagnosis of WD. AI will probably be very performant in the detection of KFR, since it analyses a grey scale images in AS-OCT. Indeed, newest program of AI dedicating to the analysis of the anterior segment by AS-OCT, notably the cornea, are in development [69–71]. The difficulty is to recruit and perform this examination in a sufficient number of patients to train the AI with a good performance. The other challenge would be the realization of good images since some patients have neurologic movements making difficult this achievement. In summary, the association of more sensitive non-dependent operator technics and the development of AI could improve drastically the diagnosis of WD reducing the diagnostic time. “

References added are the following ones:

[2] Roberts, E.A.; Schilsky, M.L. Diagnosis and Treatment of Wilson Disease: An Update. Hepatology 2008, 47, 2089–2111, doi:10.1002/hep.22261.

[68] Ferenci, P.; Caca, K.; Loudianos, G.; Mieli-Vergani, G.; Tanner, S.; Sternlieb, I.; Schilsky, M.; Cox, D.; Berr, F. Diagnosis and Phenotypic Classification of Wilson Disease 1: Diagnosis and Phenotypic Classification of Wilson Disease. Liver International 2003, 23, 139–142, doi:10.1034/j.1600-0676.2003.00824.x.

[29] Socha, P.; Janczyk, W.; Dhawan, A.; Baumann, U.; D’Antiga, L.; Tanner, S.; Iorio, R.; Vajro, P.; Houwen, R.; Fischler, B.; et al. Wilson’s Disease in Children: A Position Paper by the Hepatology Committee of the European Society for Paediatric Gastroenterology, Hepatology and Nutrition. Journal of Pediatric Gastroenterology & Nutrition 2018, 66, 334–344, doi:10.1097/MPG.0000000000001787.

[69] Ting, D.S.J.; Foo, V.H.; Yang, L.W.Y.; Sia, J.T.; Ang, M.; Lin, H.; Chodosh, J.; Mehta, J.S.; Ting, D.S.W. Artificial Intelligence for Anterior Segment Diseases: Emerging Applications in Ophthalmology. Br J Ophthalmol 2021, 105, 158–168, doi:10.1136/bjophthalmol-2019-315651.

[70] Elsawy, A.; Eleiwa, T.; Chase, C.; Ozcan, E.; Tolba, M.; Feuer, W.; Abdel-Mottaleb, M.; Abou Shousha, M. Multidisease Deep Learning Neural Network for the Diagnosis of Corneal Diseases. American Journal of Ophthalmology 2021, 226, 252–261, doi:10.1016/j.ajo.2021.01.018.

[71] Dahrouj, M.; Miller, J.B. Artificial Intelligence (AI) and Retinal Optical Coherence Tomography (OCT). Seminars in Ophthalmology 2021, 36, 341–345, doi:10.1080/08820538.2021.1901123.

We would like to thank you again for your interest and thorough comments. We remain at your disposal for more detailed and additional reviews, should you consider this necessary.

Respectfully yours,

Reviewer 2 Report

In this systematic review, the authors analyzed the most recent data on the eye involvement in Wilson's disease patients based on 49 published papers. The paper is well written, including new data, interesting figures (Figure 4 may be enlarged for a better view), and describing new methods for analyzing the eye involvement in WD. 

 I wonder if the authors did find differences in the incidence of KFR in the pediatric populations and adult studies. It would be interesting to add this figure.

Also, I would move the idea of AI in the analysis of ophthalmic imaging from conclusions to discussions, and I would develop it.  

Author Response

Dear Reviewer 2,

First, we would like to thank you for the time and effort that you dedicated with your feedback on our manuscript. We are grateful for the insightful comments that add valuable improvements to our paper. We have incorporated most of your suggestions. Those changes are highlighted within the manuscript. Please see below, in blue, a point-by-point response to your comments.

  1. Figure 4 may be enlarged for a better view

Thank you. We have enlarged figure 4 as per your request.

  1. I wonder if the authors did find differences in the incidence of KFR in the pediatric populations and adult studies. It would be interesting to add this figure.

We totally agree with this pertinent comment. A very recent description of 182 children’s cohort has been published. This is the largest pediatric cohort in WD. In this study, at diagnosis, 149 (81.8%) children had an ophthalmologic evaluation. Among them, 58 (38.9%) had a detectable Kayser-Fleischer ring (KFR): 40/129 (31.0%) were hepatic patients and 18/19 (94.7%) were neurological patients. The youngest patient with a detectable Kayser Fleischer ring was 7-year old, and a total of 8 patients (13.7%), with detectable Kayser Fleischer rings were younger than 10 years, all were hepatic patients. It’s puzzling that KFR is more frequent in neurological patients (like in adults) but more early in hepatic patients. To compare, in the largest adult cohorts from Merle et al. (163 patients), KFR was detected in 66.3% of the patients and more frequently in those with neurological symptoms than those with hepatic symptoms (85.5% vs 52.1%, p < 0.001). To resume this comment, we added a section in the results along with a Table (Table 1): “The incidence of KFR also varies according to the age of the diagnosis. Indeed, the largest pediatric cohort in WD have been recently published by Couchonnal et al. describing 182 children with WD. In this cohort, at diagnosis, 149 (81.8%) children had an ophthalmologic evaluation [24]. Among them, 58 (38.9%) had a detectable KFR: 40/129 (31.0%) were hepatic patients and 18/19 (94.7%) were neurological patients. The youngest patient with a detectable KFR was 7-year-old, and a total of 8 patients (13.7%), with detectable KFR were younger than 10 years, all were hepatic patients. The incidence of KFR is so lower than adult. Nevertheless, it is puzzling that KFR is more frequent in neurological patient (like in adults) but more early in hepatic patients. To compare, in one of the largest adult cohorts from Merle et al. (163 patients), KFR was detected in 66.3% of the patients and more frequently in those with neurological symptoms than those with hepatic symptoms (85.5% vs 52.1%, p <0.001) [25] (Table 1).”

References added are the following ones:

[24] Couchonnal, E.; Lion-François, L.; Guillaud, O.; Habes, D.; Debray, D.; Lamireau, T.; Broué, P.; Fabre, A.; Vanlemmens, C.; Sobesky, R.; et al. Pediatric Wilson’s Disease: Phenotypic, Genetic Characterization and Outcome of 182 Children in France. Journal of Pediatric Gastroenterology & Nutrition 2021, 73, e80–e86, doi:10.1097/MPG.0000000000003196.

[25] Merle, U.; Schaefer, M.; Ferenci, P.; Stremmel, W. Clinical Presentation, Diagnosis and Long-Term Outcome of Wilson’s Disease: A Cohort Study. Gut 2007, 56, 115–120, doi:10.1136/gut.2005.087262.

Table 1 was added:

Table 1 – Incidence of Kayser Fleisher Ring according to the form of the disease and the population

Pediatric population [24]

Adult population [25]

Patients undergoing SLE

149

163

Neurological patients

19

55

Hepatic patients

129

96

Incidence of KFR

58 (38.9%)

108 (66.3%)

Neurological patients

18 (94.7%)

47 (85.5%)

Hepatic patients

40 (31.0%)

50 (52.1%)

KFR: Kayser Fleisher Ring, SLE: Slit Lamp Examination

  1. Also, I would move the idea of AI in the analysis of ophthalmic imaging from conclusions to discussions, and I would develop it.

We would like to thank you for this comment. As per your request,  we added theses sentences in the discussion: “Finally, the development of artificial intelligence (AI) programs such anterior segment images analysis could be a huge step in the diagnosis of WD. AI will probably be very performant in the detection of KFR, since it analyses a grey scale images in AS-OCT. Indeed, newest program of AI dedicated to the analysis of the anterior segment by AS-OCT, notably the cornea, are in development [69–71]. The difficulty is to recruit and perform this examination in a sufficient number of patients to train the AI with a good performance. The other challenge would be the realization of good images since some patients have neurologic movements making difficult their achievement. In summary, the association of more sensitive non-dependent operator techniques and the development of AI could drastically improve the diagnosis of WD reducing the diagnostic time.”

References added are the following ones:

[67] Ting, D.S.J.; Foo, V.H.; Yang, L.W.Y.; Sia, J.T.; Ang, M.; Lin, H.; Chodosh, J.; Mehta, J.S.; Ting, D.S.W. Artificial Intelligence for Anterior Segment Diseases: Emerging Applications in Ophthalmology. Br J Ophthalmol 2021, 105, 158–168, doi:10.1136/bjophthalmol-2019-315651.

[68] Elsawy, A.; Eleiwa, T.; Chase, C.; Ozcan, E.; Tolba, M.; Feuer, W.; Abdel-Mottaleb, M.; Abou Shousha, M. Multidisease Deep Learning Neural Network for the Diagnosis of Corneal Diseases. American Journal of Ophthalmology 2021, 226, 252–261, doi:10.1016/j.ajo.2021.01.018.

[69] Dahrouj, M.; Miller, J.B. Artificial Intelligence (AI) and Retinal Optical Coherence Tomography (OCT). Seminars in Ophthalmology 2021, 36, 341–345, doi:10.1080/08820538.2021.1901123.

We would like to thank you again for your interest and thorough comments. We remain at your disposal for more detailed and additional reviews, should you consider this necessary.

Respectfully yours,

Round 2

Reviewer 1 Report

Accept in present form